Neural correlates of tactile hardness intensity perception during active grasping

Kim Ji-Hyun 1
Kim Junsuk 2
Yeon Jiwon 3
Park Jang-Yeon 4 5
http://orcid.org/0000-0003-1999-0326 Chung Dongil 1 dchung@unist.ac.kr
Kim Sung-Phil 1 spkim@unist.ac.kr
1 Department of Biomedical Engineering, Ulsan National Institute of Science and Technology , Ulsan , Republic of Korea
2 Department of Industrial ICT Engineering, Dong Eui University , Busan , Republic of Korea
3 Department of Psychology, Georgia Institute of Technology , Atlanta, Georgia , United States
4 Department of Biomedical Engineering, Sungkyunkwan University , Suwon , Republic of Korea
5 Department of Intelligent Precision Healthcare Convergence, Sungkyunkwan University , Suwon , Republic of Korea
Abdullah Jafri
Electronic publication date: 2021 Aug 2
Publication date: 2021
Volume: 9
Electronic Location ID: e11760
Received 2021 Mar 5; Accepted 2021 Jun 21
Copyright: © 2021 Kim et al.
Copyright year: 2021
Copyright holder: Kim et al.
License: This is an open access article distributed under the terms of the Creative Commons Attribution License, which permits unrestricted use, distribution, reproduction and adaptation in any medium and for any purpose provided that it is properly attributed. For attribution, the original author(s), title, publication source (PeerJ) and either DOI or URL of the article must be cited.
License URL: https://creativecommons.org/licenses/by/4.0/

Keywords: fMRI, GLM, Active grasping, Brain, Hardness, Tactile

Funding: National Research Foundation NRF-2018-Global Ph.D. Fellowship Program U-K (UNIST-Korea) 1.210046.01 National Research Foundation NRF-2019M3E5D2A01058328 This work was supported by an NRF Grant funded by the Korean Government (NRF-2018-Global Ph.D. Fellowship Program), the U-K (UNIST-Ulsan, South Korea) research brand program (1.210046.01) funded by UNIST (Ulsan National Institute of Science & Technology, Ulsan, South Korea) and the Brain Convergence Research Program of the National Research Foundation (NRF) funded by the Korean government (MSIT) (NRF-2019M3E5D2A01058328). The funders had no role in study design, data collection and analysis, decision to publish, or preparation of the manuscript.

==============================
While tactile sensation plays an essential role in interactions with the surroundings, relatively little is known about the neural processes involved in the perception of tactile information. In particular, it remains unclear how different intensities of tactile hardness are represented in the human brain during object manipulation. This study aims to investigate neural responses to various levels of tactile hardness using functional magnetic resonance imaging while people grasp objects to perceive hardness intensity. We used four items with different hardness levels but otherwise identical in shape and texture. A total of Twenty-five healthy volunteers participated in this study. Before scanning, participants performed a behavioral task in which they received a pair of stimuli and they were to report the perceived difference of hardness between them. During scanning, without any visual information, they were randomly given one of the four objects and asked to grasp it. We found significant blood oxygen-level-dependent (BOLD) responses in the posterior insula in the right hemisphere (rpIns) and the right posterior lobe of the cerebellum (rpCerebellum), which parametrically tracked hardness intensity. These responses were supported by BOLD signal changes in the rpCerebellum and rpIns correlating with tactile hardness intensity. Multidimensional scaling analysis showed similar representations of hardness intensity among physical, perceptual, and neural information. Our findings demonstrate the engagement of the rpCerebellum and rpIns in perceiving tactile hardness intensity during active object manipulation.

Introduction

Tactile sensation with the hands is essential for humans to manipulate objects in their daily lives. When manipulating objects with various shapes and textures, humans sense the objects’ physical properties via tactile information processing, beginning with interactions between sensory receptors underneath the skin and the object (Skedung et al., 2013). Not only does tactile sensation play an important role in the perception of an object’s physical properties but it also facilitates precise object manipulation through sensory feedback (Augurelle et al., 2002; Monzée, Lammarre & Smith, 2003).

Much effort has been devoted to elucidating the fundamental elements of tactile perception. By and large, human tactile perception can be characterized by four fundamental perceptual dimensions: roughness/smoothness, stickiness/slipperiness, warmth/coolness, and hardness/softness (Hollins et al., 2000; Bensmaïa & Hollins, 2005). As for tactile roughness perception, the spatial patterns of skin deformation play a key role in the discrimination of roughness (Lederman & Taylor, 1972). In particular, simulated temporal cues (Cascio & Sathian, 2001; Gamzu & Ahissar, 2001) as well as tangential forces are known to be important in determining roughness intensity (Smith et al., 2002). As for stickiness perception, its magnitude closely correlates with a parameter of kinetic friction between the skin and a surface (Smith & Scott, 1996). The perceived magnitude of warmth or coolness is determined by contact coefficients and the heat conducted on the skin area (Ho & Jones, 2006; Ho & Jones, 2008; Stevens & Marks, 1979). Lastly, as for hardness perception, both tactile and kinesthetic information are necessary to discriminate hardness levels (Srinivasan & LaMotte, 1995).

A number of functional magnetic resonance imaging (fMRI) studies have explored tactile information processing in the human brain. Several brain regions including the primary somatosensory area (SI), supplementary motor area, and bilateral temporal poles (Kim et al., 2015) are known to be involved in the perception of tactile roughness. In addition, the secondary somatosensory area (SII) was found to be activated with high-frequency stimulus (Seri et al., 2019; Seri et al., 2020), while the parietal operculum lobe is reportedly engaged in the estimation of surface roughness (Kitada et al., 2005; Stilla & Sathian, 2008), and the posterior insula shows haptic selectivity in processing visuo-haptic stimuli (Stilla & Sathian, 2008). With respect to stickiness, neural responses to sticky stimuli were found in the contralateral SI and ipsilateral dorsolateral prefrontal cortex (Yeon et al., 2017), and particularly those in the posterior parietal regions were found to discriminate stickiness intensity (Kim et al., 2017). Furthermore, neural activity in the dorsal posterior insula is correlated with thermal sensation of low temperature (Craig et al., 2000, Peltz et al., 2011; Oi et al., 2017). Recently, it was confirmed that the tactile network, including SI and SII, was activated during detection of tactile gratings using an advanced tactile stimulus that makes a grating using electro-static friction (Vuong et al., 2020).

Among all tactile perceptual dimensions, this study aims to investigate neural activity related to hardness. Because tactile hardness is defined by the ratio between the relative force exerted on the surface and its displacement, one can posit that tactile hardness perception relies on proprioception (Okamoto, Nagano & Yamada, 2013). However, several studies have shown that tactile cues are more attributed to the perception of cutaneous hardness than proprioception (Srinivasan & LaMotte, 1995; Bergmann Tiest, 2010). As for the neural substrates of tactile hardness perception, a non-human primate study showed that surface compliance modulated the activity of slowly adapting type-1 afferents, but not rapidly adapting afferents (Srinivasan & LaMotte, 1995). Other studies in non-human primates revealed that neural activities in areas 3b and 1 in SI could discriminate hard and soft objects (Randolph & Semmes, 1974) and that lesions in the SII decreased discrimination performance between hard and soft stimuli (Murray & Mishkin, 1984). In humans, a few studies have shown the involvement of the SI and the parietal operculum, such that cerebral blood volume in these regions increased during hardness sensation. Moreover, the neural activity from the parietal operculum predicts different levels of object softness (Bodegård et al., 2003) during passive touch (Kitada et al., 2019). To date, converging evidence suggests the involvement of the bilateral parietal operculum in perceiving tactile hardness along with other sensory information (Lederman et al., 2001; Reed, Shoham & Halgren, 2004; Kim et al., 2019).

However, it is still unknown how tactile hardness intensity information is processed in the human brain during active exploration. Hence, this study aimed to investigate neural representations of tactile hardness intensity while grasping an object with the hand. In order to recognize an object’s hardness, both cutaneous tactile cues and dexterous manipulation applying pressure to an object are needed (Bergmann Tiest, 2010). This active exploration process is difficult to control but nevertheless an important factor in hardness perception.

Previous studies have shown that pressing or grasping an object is needed to perceive its hardness (Harper & Stevens, 1964; Turvey & Carello, 1995) and that active exploration could improve identification of the tactile properties of an object over passive exploration (Gibson, 1962). As such, we expected a more natural identification of the neural correlates of the perception of tactile hardness during active tactile exploration.

The hypothesis of this study is that SI and SII neural responses would represent hardness intensities during active grasp exploration. To this end, we employed a dexterous grasping task to elicit hardness sensation (Lederman et al., 2001) and attempted to identify the brain regions that track tactile hardness intensity. Participants performed both psychophysical and fMRI experiments: the psychophysical experiment was designed to measure perceived differences in tactile hardness intensities between stimuli, and the fMRI experiment was designed to measure the neural responses to different hardness intensities.

Materials & methods

Participants and ethics approval

A total of twenty-five healthy volunteers (mean age = 24 ± 4 years old, age range = 19–29, nine male and 16 female), with no contraindications against MRI and no history of neurological disorders, participated in this study. All participants were right-handed and had no deficits in tactile processing. Experimental procedures were approved by the ethics committee of the Ulsan National Institute of Science and Technology (UNISTIRB-15-16-A). The study was conducted in accordance with the Declaration of Helsinki. All participants were informed of the study objectives and experimental procedures and voluntarily submitted a written consent form.

Tactile stimuli

The tactile stimuli for the current study had the same physical properties (e.g., shape and surface texture) except their hardness intensity. To satisfy this criterion, we selected a commercially available product (Eggsercizer, Magister Corp.; Chattanooga, TN, USA), primarily used in motor rehabilitation (e.g., grip strengthening). The Eggsercizer is made of synthetic rubber and silicon and has an oval shape with a diameter of 5.1 cm and a height of 7 cm (Figs. 1A and 1B). The stimulus weights were approximately identical (75.24 ± 0.4 g). The physical hardness intensities were quantitatively measured by resistance to indentation using a Durometer Hardness OO-Type device (ASKER/Shore; OO-Type) at KOPTRI (Korea Polymer Testing & Research Institute, Seoul, South Korea). For the experiment, four different stimuli with different hardness intensities (28, 36, 45, and 57 a.u.) were prepared where a higher number indicated a harder stimulus (physical hardness intensity of each stimulus is denoted as H28, H36, H45, and H57, hereafter). Although these intensity values did not map exactly to perceived hardness, in general, tactile hardness intensities of 20 and 50 were reportedly perceived similar to those of chewing gum and solid truck tires, respectively. The stimuli were kept clean after each experiment using a wet tissue until their surface was free of matter.

Figure 1 Stimuli and behavioral experiment procedure.

(A) Grasping posture during experiment. (B) Each stimulus featured one of the four hardness intensities measured by the durometer: 28, 36, 45, and 57. The stimuli used in this study are commercially available (Eggsercizer, Magister Corp.; Chattanooga, TN, USA). The stimuli were made of synthetic rubbers. Each stimulus had an identical oval shape with a 5.1-cm diameter and 7-cm height. (C) The behavioral task was conducted before the fMRI experiment. Participants were asked to report perceived hardness differences between each provided pair of stimuli. A set of ten pairs of stimuli was provided to participants, which was repeated four times.

Experimental procedure

Before the fMRI experiment, participants performed a behavioral task in which they received a pair of stimuli with eyes closed (Fig. 1C). During the behavioral experiment, participants sat on a chair and put their right hands on a desk in front of the chair. To eliminate any visual effects, participants were blindfolded during the entire behavioral experiment. An experimenter placed the stimulus on the palm of the participants’ right hand with the thin side oriented toward the thumb. The stimulus was placed at the palm center with the major axis perpendicular to the fingers except the thumb and participants grasped each stimulus once for 1 s and released it. Participants were instructed to exert a grasping force as constantly as possible for every stimulus. Immediately after participants released the stimulus, the experimenter removed it from participants’ hands and placed the next stimulus after a 5-s inter-stimulus interval. In the instruction session, an experimenter presented to participants a pair of stimuli with the highest (H57) and lowest intensities (H28) to allow participants to perceive the greatest difference in the hardness intensity of the experimental stimuli. Then, a pair of stimuli with the same hardness intensity (randomly selected out of four levels) was presented to participants along with information about the same hardness intensity.

Participants performed four main sessions of the behavioral task in each of which a total of 10 stimulus pairs were presented to participants. This set of 10 pairs consisted of pairs of stimuli with different hardness and four additional pairs with the same hardness level (4C2 + 4 = 10 pairs). In each trial of the behavioral task, participants were presented with one pair of stimuli after the other and grasped each stimulus as described above. Then, they verbally reported a perceived difference. Each participant was allowed to report a difference in their own numerical scale, which would be normalized in the following data analyses (see Data analysis for normalization details). Each main session began with the instruction session. There was a maximum 60-s break between successive sessions. Overall, a total of 40 pairs (10 pairs × 4 repetitions) were presented to participants during the behavioral experiment.

The objective of the fMRI experiment was to identify brain regions related to tactile hardness perception and neural activities representing the hardness intensity during active exploration of presented objects. As in the behavioral experiment, stimuli with four hardness levels were presented to participants. In addition, participants were given a “sham” stimulus as a control task where they executed the same grasping and releasing motions without holding any objects. During this control task, participants were asked to curl their fingers as if they grasped a virtual object in their hands, following the visual cue. Note that participants were instructed to avoid contacting their fingers with the palm to minimize the potential effects of unnecessary tactile sensations.

During the scanning sessions, participants laid comfortably on the MRI bed, leaving their right palm facing up. A total of two or three foam cushions were placed in the space between the head and the head coil to fix the participant’s head, minimizing movement effects. Participants were instructed to grasp the stimuli with constant force, and to focus on stimuli’s hardness during the grasping action. The procedure of each trial is illustrated in Fig. 2. An experimenter stood at the entrance of the magnet bore and presented stimuli along the screen’s instruction. After a 15-s baseline period, participants performed the grasping task by following the instructions shown on a visual display. Each trial comprised five 3-s grasping periods, taking 15 s total (Fig. 2). In each grasping period, the cue “Grab” was presented for 2 s. At the same time, the experimenter placed a stimulus on participants’ hands. Each time participants viewed the ‘Grab’ instruction, they were asked to grasp the stimulus for 1 s and released it afterwards. After 15 s, the stimulus was collected by the experimenter and participants held a resting position for 9 s until the next trial. The same stimulus was grasped repeatedly by participants in a single trial. A total of five stimuli (four hardness levels plus one sham stimulus) were presented five times each in a fully randomized order over 25 trials per session. Participants conducted two sessions overall. Each session took 618 s on average.

Figure 2 fMRI experiment process.

The fMRI task consisted of 25 active grasping and resting trials in an interleaved order; each grasping trial comprised five 3-s grasping periods where participants were asked to grab each stimulus for 1 s and release it for the subsequent 2 s. In each grasping period, participants were instructed to follow the visual cues of instructions presented on the screen.

MRI Data acquisition and preprocessing

The blood oxygen-level-dependent (BOLD) signal of participants during the tactile hardness perception task was measured using a 3-T fMRI scanner (Magnetom TrioTim, Siemens Healthineers; Erlangen, Germany). Three-dimensional (3D) functional images constructed from 48 slices covered the whole cerebrum (T2*-weighted gradient echo planar imaging, covering the whole depth of brain area, repetition time [TR] = 3,000 ms, echo time [TE] = 30 ms, flip angle = 90°, field of view [FOV] = 192 mm, slice thickness = 3 mm, and voxel size = 2.0 × 2.0 × 3.0 mm3). Anatomical high-resolution images were also acquired (T1-weighted 3D magnetization prepared rapid gradient echo sequence, TR = 2,300 ms, TE = 2.28 ms, flip angle = 8°, FOV = 256 mm, voxel size = 1.0 × 1.0 × 1.0 mm3). Functional images were preprocessed using SPM12 software (Wellcome Department of Imaging Neuroscience; London, UK), following the sequence: slice-timing correction, realignment, co-registration, segmentation, spatial normalization to the Montreal Neurological Institute (MNI) template and smoothing with a 6-mm full-width-half-maximum isotropic Gaussian kernel.

Data analysis

We excluded four participants from all data analyses because of excessive head movements over the maximal translation (2, 2, and 3 mm in x, y, z, respectively). Consequently, behavioral and functional data analyses were performed on 21 participants. To estimate the perceptual magnitude of hardness intensity, participants’ responses were first normalized into a scale from 0 to 10 using the unity-based normalization given by

(1) x′=x−min(x)max(x)−min(x)×10

where x is the original value of participants’ responses and x′ is the normalized value between 0 to 10. This normalization process was performed on each participant’s data.

Then, we examined a possible order effect of stimulus presentation within stimulus pairs (see Behavioral experiment). Specifically, we used a paired t-test to compare the normalized responses between the trials where the same set of stimuli were presented in the opposite order (e.g., [H28, H57] vs. [H57, H28]). In addition, we used a repeated-measures analysis of variance (rmANOVA) to compare the normalized responses among trials involving different levels of tactile hardness.

We performed event-related fMRI analyses of participants’ BOLD responses to tactile hardness perception. Two separate design matrices were used for first-level general linear model (GLM) analyses: one design matrix to find brain regions involved in tactile hardness perception, and the other to find the brain regions that track hardness intensity. In the first design matrix, all stimulus types (four hardness levels plus one sham) were modeled as separate linear regressors. We specified tactile perception events as the time at which participants grasped the stimuli with their hand. All events in each regressor were convolved with the canonical hemodynamic response function. At first-level, contrast images that reflected BOLD response differences between all actual versus sham tactile perceptions were generated for each participant. The second-level model was constructed as a one-sample t-test using the contrast images from the first-level. Where indicated, family-wise error (FWE) was controlled to correct for multiple comparisons.

In the second design matrix, the subjective hardness intensity obtained from the behavioral experiment was added as a parametric modulator to identify the brain regions that track tactile hardness intensity. At first-level, contrast images that reflected whole-brain responses positively and negatively correlated with hardness intensity were generated for each participant. The same strategy as with the first design matrix (contrast analysis described above) was followed for second-level analysis and multiple comparisons correction. For both contrast and parametric modulation analyses, we labeled statistically significant clusters using the automated anatomical labeling toolbox (Tzourio-Mazoyer et al., 2002).

Furthermore, we calculated and depicted the percent signal change (PSC) within the brain regions identified by parametric modulation analysis to better illustrate the region’s responses tracking the stimuli hardness information. PSC was calculated as the BOLD signal ratio in response to stimuli over that without stimulus. This analysis aimed to supplement the finding of brain regions above by confirming the correlation between the PSC in the identified regions and hardness intensity. We set the activated regions determined by the parametric modulation analysis as regions of interest (ROIs) and estimated the PSC of each single ROI in response to each hardness stimulus using the Marsbar toolbox (Brett et al., 2016).

To examine relationships between physical intensities, perceptual responses, and neural responses regarding tactile hardness, we further constructed the dissimilarity matrices of each data type. The PSC in the ROIs above was used as representational neural responses to tactile hardness. As the perceptual responses were normalized in a scale between 0 and 10, neural and physical intensity data were similarly normalized between 0 and 10. A dissimilarity matrix of physical hardness was constructed using the normalized differences between every pair of the four tactile stimuli, resulting in a 4 × 4 symmetric matrix with diagonal entries equal to zero. We constructed the dissimilarity matrix of perceptual responses in the following manner. First, participants’ perceptual responses to each stimulus were normalized within a scale from 0 to 10 using the unity-based normalization given in Eq. (1). Second, the difference of the normalized perceptual responses between every possible pair of stimuli was calculated. Third, the difference values for each pair were averaged over all participants. Fourth, a dissimilarity matrix of perceptual responses was constructed from each averaged difference. The dissimilarity matrix of neural responses of each ROI was also constructed using the normalized PSC difference values averaged across participants but resulting in a 4 × 4 symmetric matrix.

A Mantel test, often used for analyzing functional connectivity in the brain (Glerean et al., 2016), was employed to find statistically significant correlations (Pearson’s correlation) between dissimilarity matrices (Mantel, 1967).

Lastly, we applied multidimensional scaling (MDS) to the dissimilarity matrices to find low-dimensional representations of physical, perceptual, and neural responses. These low-dimensional representations of hardness intensity were meant to visualize similarities between the three different domains of tactile hardness: physical, perceptual, and neural representation of object hardness.

Results

Behavioral responses to tactile stimuli with different hardness intensities

We first confirmed the lack of order effects in perceived difference between the stimuli set. Thus, we did not further consider presentation order in subsequent analyses of behavioral responses. It also led us to define the absolute perceptual differences between a pair of stimuli as the perceived distance of tactile hardness. Then, we examined differences in perceived distance among all possible pairs of tactile stimuli using rmANOVA and observed a significant difference of mean perceived distance among those pairs (F ((9,180)) = 144.62, p < 0.0001) (Fig. 3). A post-hoc Bonferroni test further revealed significant differences in perceived distance between pairs with dissimilar hardness differences (all p < 0.0001; Fig. 3) but no difference between pairs with similar hardness differences. As an additional explorative analysis, we examined potential gender differences in the perceived distance among all possible pairs of tactile stimuli using two sample t-tests, but none of the pairs showed significant difference between male and female participants (Table S1). These results show that participants could accurately perceive objective differences in hardness between pairs of tactile stimuli.

Figure 3 Behavioral experiment results.

Participants were able to perceive the objective hardness level differences between the paired stimuli. Error bars indicate s.e.m. ***p < 0.001; n.s., not significant.

Neural responses to tactile stimuli with different hardness intensities

We examined the BOLD signal evoked by tactile hardness by contrasting each hardness intensity against the “Sham” using GLM analysis. However, we found no commonly activated regions across the whole brain. To examine the brain regions that track the physical properties (hardness intensity) of stimuli, we ran a whole-brain analysis using a design matrix including physical hardness intensity as a parametric modulator. We found a significant positive correlation between hardness intensity and neural responses in the posterior insula in the right hemisphere (rpIns) (MNI space coordinates of x = 40, y = −26, z = 18; FWE corrected p < 0.05, cluster size > 10, cluster defining threshold p < 0.001) (Fig. 4A). A set of brain regions negatively tracked physical hardness intensity (Fig. 5A). And we found a significant negative correlation in the right posterior lobe of the cerebellum (rpCerebellum) (MNI space coordinates of x = 42, y = −74, z = −40; FWE corrected p < 0.05, cluster size > 10, cluster defining threshold p < 0.001) (Fig. 5A).

Figure 4 Neural responses in the posterior insula in the right hemisphere (rpIns) positively track physical hardness intensity.

(A) A network of brain regions including rpIns positively tracked perceptual hardness intensity of stimuli (T activation map from parametric modulation, family-wise error p < 0.05; displayed at p < 0.001, cluster size > 10 for illustrative purposes). (B) For illustrative purposes, we estimated percent signal change (PSC) in the rpIns region of interest (ROI). PSC shows that the blood oxygen-level dependent responses in the ROI positively tracked the physical hardness intensity of stimuli. There were a set of brain regions that negatively tracked physical hardness intensity (Fig. 4A). We found a significant negative correlation in the right posterior lobe of the cerebellum (rpCerebellum) (MNI space coordinates of x = 42, y = −74, z = −40; FWE corrected p < 0.05, cluster size > 10, cluster defining threshold p < 0.001) (Fig. 4A).

Figure 5 Neural responses in the right posterior lobe of the cerebellum (rpCerebellum) negatively track physical hardness intensity.

(A) rpCerebellum negatively tracked perceptual hardness intensity of stimuli (T activation map from parametric modulation, family-wise error p < 0.05; displayed at p < 0.001, cluster size > 10 for illustrative purposes). (B) For illustrative purposes, we estimated percent signal change (PSC) in the cerebellum region of interest (ROI). PSC shows that the blood oxygen-level dependent responses in the ROI negatively tracked the physical hardness intensity of stimuli.

Dissimilarity analysis

We constructed three dissimilarity matrices of physical hardness intensities, perceived responses, and neural responses to hardness intensities (Fig. 6). For the neural dissimilarity matrix, PSC differences between the rpIns and rpCerebellum were used as a neural representation of tactile hardness.

Figure 6 Dissimilarity matrix of the stimuli based on physical, perceptual hardness, and neural data.

Measurements of physical, perceptual, and neural differences between all possible pairs of four stimuli (H28, H36, H45 and H57) produced three symmetric dissimilarity matrices (see text for difference measures). The color bar illustrates differences in the normalized scale from 0 to 10.

MDS analysis of the dissimilarity matrices produced two-dimensional (2D) representations of physical, neural, and perceptual hardness intensity (Fig. 7). This 2D depiction clearly visualized the response clustering into four groups according to hardness intensity across all three types of representations (physical, perceptual, and neural). Moreover, it appeared that only one dimension (MDS1) was sufficient to effectively represent hardness intensity, not only for physical stimuli properties but also for neural and perceptual responses.

Figure 7 Multidimensional scaling results.

The relative positions of the four different types of hardness intensities across physical, perceptual, and neural properties are depicted in two-dimensional space.

Discussion

The aim of the current study was to investigate neural responses to various levels of tactile hardness during active object grasping. The behavioral results clearly demonstrated that each stimulus could be distinguished by its physical hardness intensity. We found that the neural responses in the rpIns positively track the perceived tactile hardness intensity, while those of the rpCerebellum negatively track hardness intensity. The low-dimensional representations of physical hardness, perceptual responses, and neural responses obtained by MDS analysis, commonly revealed a clear segregation of hardness intensity, again supporting that the rpIns and rpCerebellum encode tactile hardness intensity information. To the best of our knowledge, this is the first study to reveal the neural correlates of hardness intensity perception during active tactile exploration in humans.

Perceptual discrimination of tactile hardness

The behavioral results showed that participants could distinguish different levels of tactile hardness when they grasped the presented stimuli. Self-reported perceptual differences between stimuli well-reflected the relative physical difference of objects’ hardness. In our data, stimulus presentation order did not affect the recognition of hardness differences. This might be due to the fact that we only used four different types of stimuli (i.e., the four stimuli resulted in six distinct pairs); participants might have easily learned the minimum and maximum hardness at the beginning of each session which might helped with accurate estimation. Nonetheless, we prepared stimuli with clearly separated hardness for this study because we wanted to explore neural substrates of well explained tactile perception.

Brain regions representing tactile hardness intensity

By comparing brain responses to stimuli against sham, we attempted to find brain regions involved in hardness intensity perception. Because all stimuli had the same shape and surface texture, we expected to identify the brain regions commonly involved in different hardness intensity perception by contrasting each hardness intensity against the sham trials. However, there was no such commonly responsive brain region, including the SI or SII. A possibility is that the SI might have been involved in sham trials due to participants’ imaginary tactile perception. Previous rodent studies showed similar prominent activation in both the MI and SI during motor control (Diamond et al., 2008; Matyas et al., 2010). In our task, hand movements were dominant in all trials because participants perceived hardness via active exploration, leading to SI activations in response to both sham and tactile stimulations.

We found that neural responses in the rpIns increased as a function of perceived tactile hardness. Conversely, neural activity in the rpCerebellum decreased with increased perceived tactile hardness. The rpIns has been reported to be involved in tactile pressure and proprioceptive perception. Previous studies reported bilateral posterior insula activation during a tactile hardness perception task (Lederman et al., 2001) and the involvement of the posterior insula in tactile and body displacement perception and bodily self-consciousness (Richer et al., 1993; Blefari et al., 2017; Findlater et al., 2018), as well as activation in response to tactile pressure stimulation (Chung et al., 2015). Based on previous lesion studies, the posterior insula is known to be involved in self-awareness of actions (Karnath & Baier, 2010). Previous research backs up our finding that rpIns might be engaged in the perception of hardness intensity during active grasping. Our results provide additional evidence that the human posterior insula might play a role in the tactile perception of hardness intensity. While we found activation only in the rpIns, a more liberal statistical threshold (p < 0.001, uncorrected) revealed involvement of the bilateral posterior insula. Future studies should examine the functional connectivity between the bilateral posterior insula and other brain regions, such as the SI, to understand why rpIns showed stronger responses than the contralateral hemisphere.

One of the most noticeable findings in the current study is that the cerebellum was activated during active perception of tactile hardness. A number of human neuroimaging studies has shown that the cerebellum is functionally connected to the sensorimotor network combining the sensorimotor cortex, premotor area, and supplementary motor area. In addition, the cerebellum is known to participate in motor control, motor learning, and motor planning (Habas et al., 2009; Krienen & Buckner, 2009; O’Reilly et al., 2010; Zeeuw & Brinke, 2015; Gao et al., 2018). In particular, the rpCerebellum is known to be engaged in self-produced tactile perception (Blakemore, Wolper & Frith, 1999), which may be necessary in to recognize hardness intensity. According to a recent study, functional connectivity between the primary somatosensory cortex and the cerebellum was strengthened when participants reported perceptual attenuations of self-generated touch (Kilteni & Ehrsson, 2020). These results suggest that the ipsilateral cerebellum has a crucial role in predicting self-generated touch. Our results showed that cerebellar activity is associated with perceived tactile hardness during active grasping, and are in line with the aforementioned previous findings.

Relationship between neural responses, physical, and perceptual tactile hardness

The Mantel test and MDS analysis were conducted to intuitively illustrate the relationship among physical, perceptual, and neural representations of tactile hardness intensity. The Mantel test results suggested similar perception of different levels of hardness intensity to neural responses, which were marginally correlated with the physical hardness of stimuli, suggesting that the brain tracks the subjective perception of objects’ hardness rather than their physical hardness. The MDS analysis showed similar clustering of physical, perceptual, and neural representations of each hardness level. This confirms that neural responses in both posterior insula and cerebellar regions represented perceived hardness.

Limitations

This study has a few limitations. First, we assumed that individuals show stable perceptual responses. In the behavioral experiments, we could identify the relationship between the physical and the perceived hardness for each participant. During the scanning phase, we did not directly collect participants’ subjective responses because we intended for participants to fully concentrate on the grasping motions. Under our experimental environment, it was difficult to obtain a direct response from participants during MRI scanning. During neural image acquisition, participants explored stimuli with their right hand and held a squeeze ball for emergency with their left hand, which prohibited them from making behavioral responses with button pressing. Alternatively, we attempted to receive verbal responses directly during scanning, but the scanner noise was too loud. As such, we decided to separate behavioral and neural imaging experiments while ensuring that participants maintained the same active grasping across both experiments. Therefore, the question whether the neural activations measured in the scanner would be the same as those outside remains unanswered. Similarly, albeit very unlikely, it is still unclear whether participants experienced the same hardness perception outside the scanner as inside. To minimize this difference, we trained participants to grasp the stimuli with constant force before conducting the main experiments. Future studies could use MRI-compatible electromyography (EMG) or force sensors that can measure individuals’ grasping force during the scanning phase.

Second, our findings may be limited by the fact that only four hardness intensities were used. We observed that neural responses in the cerebellum and posterior insula were correlated with perceived hardness. However, due to the small number of stimulus types, our data cannot distinguish subtle differences between models that assume linear versus non-linear (e.g., logarithmic, quadratic) associations of neural and perceptual responses.

Third, there is a chance that the neural responses measured in this study might also reflect the changing reaction force applied to the hand with hardness. To minimize this confounding effect, we intended to ensure that participants grasped the stimuli with a constant force. However, it is still impossible to determine whether participants exerted exactly the same force during the experiment. In an effort to bypass this factor, we used subjective ratings of hardness levels rather than physical hardness, so that the self-reported rates should already take the participants’ own perception of exerted force into account. Still, to further optimize the task design, future studies could use a visual feedback that shows an EMG-based gage to inform participants of their exerted force in real-time. In addition, in an experiment with real-time EMG, we could observe neural responses during tactile information processing using EEG or MEG with finer time resolution (Golnaz et al., 2020; Hagiwara et al., 2020), which would help confirm the relationship between muscle strength and neural responses in real-time.

Fourth, our experimental design solely focused on active exploration of tactile stimuli. Thus, potential individual differences in passive tactile perception may exist. Note that we controlled for such a confounding factor by examining neural responses parametrically modulated as a function of perceived stimulus hardness.

Fifth, and lastly, it is possible that the marginal association between neural activity and physical hardness could be partially due to insufficient statistical power (the sample size in this study was pre-determine based on that used in similar studies using functional neuroimaging in tactile perception (Case et al., 2016)). Our task design and instruction do not rule out the possibility that neural responses would encode physical hardness, but our results suggest that the neural responses elicited in our active grasping task are more associated with perceptual object discrimination rather than tracking of physical hardness per se.

Also, from our explorative analyses addressing potential gender differences, inconsistent with a previous report (Abdouni et al., 2018), we did not find any differences in either behavioral or neural results. Still, as this is an exploratory approach, the interpretation of this result should be cautious. Such an inconsistency could be due to lack of power, given our small sample size for each gender group, and thus, future studies would be necessary to further examine the potential effects of gender differences in tactile sensation.

Conclusions

Nevertheless, in the present study, we found several brain regions associated with hardness intensity perception during active exploring. Neural responses in the cerebellum and posterior insula regions significantly correlated with physical hardness intensity and behavioral intensity rating. Extension of the tactile dimensions (e.g., slipperiness) and exploration of a broader psychophysical range in future studies may further our understanding of human tactile information processing. Further, by using multivoxel pattern analysis and functional connectivity analyses, we can construct more sophisticated neural and behavioral models in tactile processing that may provide insights for developing realistic haptic devices and artificial intelligence that simulates human’s subjective tactile perception.

Supplemental Information

Supplemental Information 1 Explorative analysis of potential gender effects.

Results of two sample t-test of each stimuli pair. STD indicates standard deviation. H28-57: For the experiment, four different stimuli with different hardness intensities (28, 36, 45, and 57 a.u.) were prepared where a greater number indicated a harder stimulus (physical hardness intensity of each stimulus is denoted as H28, H36, H45, and H57).There was no significant difference of perceived difference of hardness in stimulus pairs by gender.

Click here for additional data file.

Supplemental Information 2 Percent signal changes for each stimulus within the ROI.

rpINS: posterior insula in the right hemisphere, rpCerebellum: right posterior lobe of the cerebellum, H28-57: For the experiment, four different stimuli with different hardness intensities (28, 36, 45, and 57 a.u.) were prepared where a greater number indicated a harder stimulus (physical hardness intensity of each stimulus is denoted as H28, H36, H45, and H57).

Click here for additional data file.

Supplemental Information 3 Participants’ responses from behavioral experiment.

The first and second columns are the stimuli given to the participant (“1” means H28 stimulus, “4” means H57 stimulus). The third column is the responses from participants regarding differences in hardness between stimuli.

Click here for additional data file.

Supplemental Information 4 Activated brain regions from parametric modulation analysis.

Click here for additional data file.

The authors would like to thank Enago for the English language review.

Additional Information and Declarations

Competing Interests

Author Contributions

Human Ethics

Data Availability

The authors declare that they have no competing interests.

Ji-Hyun Kim conceived and designed the experiments, performed the experiments, analyzed the data, prepared figures and/or tables, authored or reviewed drafts of the paper, and approved the final draft.

Junsuk Kim conceived and designed the experiments, performed the experiments, analyzed the data, authored or reviewed drafts of the paper, and approved the final draft.

Jiwon Yeon conceived and designed the experiments, analyzed the data, prepared figures and/or tables, and approved the final draft.

Jang-Yeon Park performed the experiments, authored or reviewed drafts of the paper, and approved the final draft.

Dongil Chung analyzed the data, authored or reviewed drafts of the paper, and approved the final draft.

Sung-Phil Kim conceived and designed the experiments, authored or reviewed drafts of the paper, and approved the final draft.

The following information was supplied relating to ethical approvals (i.e., approving body and any reference numbers):

Experimental procedures were approved by the ethics committee of the Ulsan National Institute of Science and Technology (UNISTIRB-15-16-A). The study was conducted in accordance with the Declaration of Helsinki.

The following information was supplied regarding data availability:

The data is available in the Supplementary Files.

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
