# Peer review of "Neural correlates of tactile hardness intensity perception during active grasping"

_PeerJ, doi:10.7717/peerj.11760_

## Round 0.1 · original submission · Major Revisions

Dear Authors,

There are major issues highlighted that needs revision. Please do this as soon as possible and resubmit for re-review.

·

Basic reporting

The manuscript is easy to understand but needs to be sent for English proofread.

Some of the literature refences are old but overall is sufficient to the background/context of the study.

I would suggest to include;

Please include flow chart or figure with description for the experiment procedure, step by step. This will make easier for reader to understand how the experiment was conducted.

Please include figure with description for the experimental paradigm.

Experimental design

2. In the abstract, the is no details on the participants, data analysis and how the tactile intensities were ratings.

3. In the method,

please also include the mean age for the female volunteers.
Twenty-five healthy volunteers (9 males, mean age: 24±4 years) participated in this study
And please also include the age range for volunteers.

Please include flow chart or figure with description for the experiment procedure, step by step. This will make easier for reader to understand how the experiment was conducted.

Please include figure with description for the experimental paradigm.

Participants were instructed to grasp the stimuli with constant force, and to focus on the hardness of the stimuli during the grasping action.
How do you control the grasp force action among the volunteers? How to make sure that all the volunteer´s force is not varies as this will contribute to brain activation intensities.

We excluded four participants from all data analyses because of excessive head movements
I am not clear here, the total participants would be 21 or, 25 participants?
21 participants for fMRI study would be under power, and we have to consider gender differences.
Abdouni et al 10.1038/s41598-018-32724-4 reported the differences between male and female processing tactile perception.
My suggestion to add more participants to have an adequate statistical power. It is difficult to draw a conclusion with only 21 participants.

Validity of the findings

The present study only have 21 participants.

21 participants for fMRI study would be under power, and we have to consider gender differences.
Abdouni et al 10.1038/s41598-018-32724-4 reported the differences between male and female processing tactile perception.
My suggestion to add more participants to have an adequate statistical power. It is difficult to draw a conclusion with only 21 participants.

Additional comments

In this study, the authors looked at that neural responses of primary or secondary somatosensory areas that would represent the hardness intensities during active grasp exploration. They found significant BOLD responses in the posterior insula in the right hemisphere and the right posterior lobe of the cerebellum, which parametrically tracked hardness intensity. This is the first study to reveal neural correlates of the perception of hardness intensity during active tactile exploration in humans. While this is an interesting topic, the current manuscript has several limitations.

1. The manuscript needs to be sent for English proofread.

2. In the abstract, the is no details on the participants, data analysis and how the tactile intensities were ratings.

3. In the method,

please also include the mean age for the female volunteers.
Twenty-five healthy volunteers (9 males, mean age: 24±4 years) participated in this study
And please also include the age range for volunteers.

Please include flow chart or figure with description for the experiment procedure, step by step. This will make easier for reader to understand how the experiment was conducted.

Please include figure with description for the experimental paradigm.

Participants were instructed to grasp the stimuli with constant force, and to focus on the hardness of the stimuli during the grasping action.
How do you control the grasp force action among the volunteers? How to make sure that all the volunteer´s force is not varies as this will contribute to brain activation intensities.

We excluded four participants from all data analyses because of excessive head movements
I am not clear here, the total participants would be 21 or, 25 participants?
21 participants for fMRI study would be under power, and we have to consider gender differences.
Abdouni et al 10.1038/s41598-018-32724-4 reported the differences between male and female processing tactile perception.
My suggestion to add more participants to have an adequate statistical power. It is difficult to draw a conclusion with only 21 participants.

4. In the discussion,
It is unclear, how author correlate between behavioural score and fMRI data and how this difference intensities of tactile correlate to behavioural data.
Authors did not describe the activation pattern according to the difference intensities of the tactile. Please also tabulate in the table the activation intensities vary in response to textile intensities.
Please also discuss in detail the activation areas evoked by this stimulus.

Thank you.

Reviewer 2 ·

Basic reporting

It is suggested to includes recent citations as references for tactile and somatosensory study in the introduction and discussion section.

Below are few recent articles that suggest being added. Please find few others in the area to support your statement in the introduction, support your findings and critically discuss it.
• Seri FAS, Abd Hamid AI, Abdullah JM, Idris Z, and Omar H (2020). Brain responses to high frequencies (270 Hz-480 Hz) changes due to vibratory stimulation of human fingertips: An fMRI study. J. Phys.: Conf. Ser. 1497 012012. doi:10.1088/1742-6596/1497/1/012012
• Seri FAS, Abd Hamid AI, Abdullah JM, Idris Z, and Omar H. (2019). Brain responses to frequency changes due to vibratory stimulation of human fingertips: An fMRI study. J. Phys.: Conf. Ser. 1248 012029. doi:10.1088/1742-6596/1248/1/012029
• Vuong QC, Shaaban AM, Black C, Smith J, Nassar M, Abozied A, Degenaar P and Al-Atabany W (2020) Detection of Simulated Tactile Gratings by Electro-Static Friction Show a Dependency on Bar Width for Blind and Sighted Observers, and Preliminary Neural Correlates in Sighted Observers. Front. Neurosci. 14:548030. doi: 10.3389/fnins.2020.548030
• Kim Y, Usui N, Miyazaki A, Haji T, Matsumoto K, Taira M, Nakamura K and Katsuyama N (2019) Cortical Regions Encoding Hardness Perception Modulated by Visual Information Identified by Functional Magnetic Resonance Imaging With Multivoxel Pattern Analysis. Front. Syst. Neurosci. 13:52. doi: 10.3389/fnsys.2019.00052

It is suggested that any tactile and somatosensory studies used other brain imaging modalities like EEG, MEG, etc., to be included in the discussion section. This will help to discuss some of your findings and limitation.

Example study in the related areas but using EEG.
• Baghdadi, Golnaz, Amiri, Mahmood, Falotico, Egidio, Laschi, Cecilia. 2020. Recurrence quantification analysis of EEG signals for tactile roughness discrimination. International Journal of Machine Learning and Cybernetics. https://doi.org/10.1007/s13042-020-01224-1

Please move the limitation in the conclusion section to the discussion part. Authors can put it under limitation and future suggestion subheading.

Experimental design

Data analysis:
1. Please include the information of acceptable head movement to be included as the participants of this study.
2. Please include the design matrix used in the first level analysis/GLM

Validity of the findings

No comment

---

## Round 0.2 · accepted · Accept

Thank you. Your Manuscript is ready for acceptance.

·

Basic reporting

The author(s) had done the amendments accordingly, and I have no further suggestions.
Thank you.

Experimental design

The author(s) had done the amendments accordingly, and I have no further suggestions.
Thank you.

Validity of the findings

The author(s) had done the amendments accordingly, and I have no further suggestions.
Thank you.

Additional comments

The author(s) had done the amendments accordingly, and I have no further suggestions.
Thank you.

Reviewer 2 ·

Basic reporting

The revised version is clear and well structured.
No comment

Experimental design

No comment

Validity of the findings

No comment

Additional comments

Overall, the authors address all questions raised by the reviewers clearer and well structured. The authors also provided few figures to better explain in visual form the paradigm and task used for this study.